# Inappropriate Use of Oral Antithrombotic Combinations in an Outpatient Setting and Associated Risks: A French Nationwide Cohort Study

**DOI:** 10.3390/jcm10112367

**Published:** 2021-05-27

**Authors:** Lorène Zerah, Dominique Bonnet-Zamponi, Aya Ajrouche, Jean-Philippe Collet, Yann De Rycke, Florence Tubach

**Affiliations:** 1Sorbonne Université, INSERM, Institut Pierre Louis d’Epidémiologie et de Santé Publique, F75013 Paris, France; 2Sorbonne Université, INSERM, Institut Pierre Louis d’Epidémiologie et de Santé Publique, AP-HP, Hôpital Pitié Salpêtrière, Département de Santé Publique, Centre de Pharmacoépidémiologie (Cephepi), CIC-1901, F75013 Paris, France; dominique.bonnet-zamponi@inserm.fr (D.B.-Z.); aya.ajrouche@aphp.fr (A.A.); yann.de-rycke@aphp.fr (Y.D.R.); florence.tubach@aphp.fr (F.T.); 3Observatoire du Médicament des Dispositifs Médicaux et de l’Innovation Thérapeutique Ile de France (OMEDIT), F-75014 Paris, France; 4Sorbonne Université, AP-HP, Hôpital Pitié-Salpêtrière, Département de Cardiologie, F75013 Paris, France; jean-philippe.collet@aphp.fr

**Keywords:** antithrombotic combinations, inappropriate prescribing, adverse drug event, hemorrhage, vascular diseases

## Abstract

With the increase in prevalence of cardiovascular diseases, multimorbidity, and medical progress, oral antithrombotic (AT) combinations are increasingly prescribed. The aims of this study were to estimate the incidence of oral AT combinations, their appropriateness (defined as indications compliant with guidelines), and the related risk of major bleeding (i.e., leading to hospitalization) or death, among new users. We conducted a 5-year historical cohort study, using the French national healthcare database, including all individuals ≥ 45 years old with a first delivery of oral ATs between 1 January 2013 and 31 December 2017. The cumulative incidence of oral AT combinations was estimated with the Fine and Gray method, taking into account the competitive risk of death. We compared the cumulative incidence of major bleeding according to the type of oral AT treatment initiated at study entry (monotherapy or oral AT combinations). During the study period, 22,220 individuals were included (mean (SD) age 68 (12) years). The cumulative incidence of oral AT combinations at 5 years was 27.8% (95% confidence interval (CI) 26.8–28.9). Overall, 64% of any oral AT combinations did not comply with guidelines. The cumulative incidence of major bleeding and death in the whole cohort at 5 years was 4.1% (95% CI 3.7–4.6) and 10.8% (95% CI 10.1–11.6), respectively. Risk of major bleeding increased among individuals with oral AT combinations versus oral AT monotherapy at study entry (subdistribution hazard ratio sHR: 2.16 (1.01–4.63)); with no difference in terms of death. The use of oral AT combinations among oral AT users is frequent, often inappropriately prescribed, and associated with an increased risk of major bleeding.

## 1. Introduction

Antithrombotics (ATs) (i.e., antiplatelet and anticoagulant therapies) are the most frequent drug class implicated in serious and fatal adverse drug events (ADEs) [1,2]. Oral AT combinations greatly increase this risk (mainly bleeding), especially when oral anticoagulation is combined with oral antiplatelet agents [3,4]. ADEs may result from medication errors that could be prevented or from adverse drug reactions not related to medication errors [5]. Previous literature estimated that 40% to 70% of ADEs are avoidable [5,6], and due to treatment misusage [7] which mainly occurs at the stage of prescribing [5].

With the increase in prevalence of cardiovascular diseases, multimorbidity and medical progress [8], oral AT combinations are increasingly prescribed. Yet, in a recent European prospective cohort study of patients with non-valvular atrial fibrillation (NV-AF) [9], 95.3% of patients on dual therapy (one oral anticoagulant [OAC] and one oral antiplatelet) and 63.8% receiving triple therapy (one OAC and 2 oral antiplatelets) had no recommended indications for use of these treatments according to guidelines, which suggested a high rate of inappropriate prescribing of oral AT combinations in patients with NV-AF. In addition, in our recent study of a sample of French physicians that used clinical vignettes illustrating cases of adult outpatients with common vascular diseases (including NV-AF, coronary artery disease, ischemic stroke, or arterial embolism, valvular heart disease, peripheral artery disease, and venous thromboembolism) [10], 76% of oral AT combination prescriptions (using OACs and/or antiplatelets) did not comply with guidelines in terms of indication, dosage or duration [10].

This current study aimed to (1) describe the patterns of oral AT combinations in French adults ≥ 45 years old who received oral ATs for all common vascular diseases in the outpatient setting, (2) assess the incidence of the use of oral AT combinations, whether prescribed in accordance with guidelines, and (3) estimate the related risks of major bleeding or death.

## 2. Materials and Methods

### 2.1. Study Design and Data Source

We conducted a 5-year historical cohort study using data from the *Échantillon Généraliste des Bénéficiaires* (EGB) (general sample of beneficiaries), a dynamic random permanent sample (1/97th) from the *Système National des Données de Santé* (SNDS, French national health care database) that includes all individuals affiliated with the French health insurance system since 2005 (general scheme: 90%) [11]. Data are stored for 20 years starting in 2005 [11].

The SNDS includes anonymous and prospectively recorded data on the demographic characteristics of beneficiaries such as date of birth, sex, vital status, and date of death. It includes some medical information such as 100% reimbursement for care of severe and costly long-term chronic disease (LTD, coded according to the International Classification of Diseases, 10th revision (ICD-10)) and all out-of-hospital health spending reimbursements [11,12]. Drugs are coded according to the Anatomical Therapeutic Chemical [ATC] classification, and the database includes information on the date of dispensation and dosage and quantity of the drug dispensed. It also contains, for each hospitalization, the hospital discharge summary (principal and related diagnoses coded according to the ICD-10) and medical procedures [11,12]. A principal diagnosis is the diagnosis established after study to be chiefly responsible for occasioning the admission to the hospital. Related diagnoses are the other conditions that are either present on admission or developed as a direct result of the principal diagnosis. The EGB includes data for over 700,000 people and has been shown to cover a population representative of the French national population for age, sex, occupation, and medical expenses [12]. A detailed description of the data source can be found elsewhere [11,12].

### 2.2. Study Population

All individuals in the EGB sample covered by the general scheme of the health insurance system were included in the current study if they were ≥ 45 years old and had a first delivery of oral AT from 1 January 2013 to 31 December 2017. The index date was the date of the first delivery of oral AT. To avoid the inclusion of prevalent AT users, individuals with AT dispensation during the year before the index date were not eligible (new user design [13]).

The study period started in 2013 when the first 2 direct oral anticoagulants (DOAs), dabigatran and rivaroxaban, arrived on the market in France. We excluded individuals with auto-immune disease, hemophilia, HIV, active cancer (see Appendix A) at the index date, as they still requiring an expert opinion for AT treatment. We focused on oral ATs and excluded individuals with a first delivery of an injectable anticoagulant (see Appendix A). We excluded individuals under the age of 45 at index date as major vascular diseases are rare in this age group, and always require an expert opinion for AT treatment.

### 2.3. Exposure

Oral ATs were identified using specific ATC codes (B01*, see Appendix A). Definition of exposure windows was based on the proportion of days covered [14,15]. Individuals were considered exposed starting on the day they filled a prescription for an oral AT drug. Length of exposure to oral ATs was estimated based on the number of pills delivered; the expected number to be taken per day; the length between deliveries, taking into account overlaps between 2 consecutive refills with the corresponding number of days carried over; and hospitalization periods during which ATs were provided by the hospital.

Exposure to oral AT combinations was defined as the delivery of at least 2 oral ATs for at least 15 successive days. This definition aimed at not considering short AT combinations corresponding to a switch of oral ATs (could not exceed 14 days according to the recommendations). Exposure to oral AT combination corresponds to the period covered by consecutive deliveries of the same oral AT combination without interruption (a line of treatment). As a sensitivity analyses, we defined exposure to oral AT combination as the delivery of at least 2 different oral ATs for at least 30 or 45 successive days.

Inappropriate oral AT combinations were defined as deliveries not complying with the guidelines, according to the synthesis of international guidelines [16], considering the medical indication for use only, but not dosage or treatment duration. It may be appropriate to prescribe outside the guidelines for some individuals or contexts, but these situations are rare and specific [16,17], and most of them were excluded from this study (requiring an expert opinion: auto-immune disease, hemophilia, HIV, or active cancer). Duration of oral AT combination use and dose of oral AT were not considered for compliance with guidelines, because the recommended duration may vary according to the bleeding risk, which cannot be precisely assessed in the database, and because creatinine clearance and weight data, required to determine the recommended dose, are not available in the database. We first considered, as inappropriate: (1) oral AT combinations always contraindicated (P2Y12 inhibitor combinations, anticoagulant combinations used longer than 15 days, dual therapy or triple therapy with ticagrelor or prasugrel and combinations of 3 antiplatelets), and (2) other oral AT combinations with no indication identified, in accordance with the guidelines. For this, for each type of oral AT combination (dual antiplatelet therapy, dual therapy [i.e., one antiplatelet and one OAC], triple therapy [i.e., 2 antiplatelets and one OAC]), we searched within the 3 (dual antiplatelet and triple therapy) or 6 months (dual therapy) before the first delivery of oral AT combinations to determine whether the individual had a recommended indication for treatment (as appropriate, see Appendix A) [16]. Events leading to the delivery of oral AT combinations are easily identifiable in the EGB, and only recent events can lead to oral AT combination prescriptions.

### 2.4. Outcomes

The primary outcome was the initiation of oral AT combinations.

Secondary outcomes were (1) the appropriateness of oral AT combinations (i.e., indication for the oral AT combination in accordance with the guidelines); (2) major bleeding, (i.e., hospitalization with bleeding as the principal diagnosis or related diagnoses), including intracranial (hospital discharge summary with ICD-10 codes I60, I61, I62, S06.3, S06.4, S06.5, S06.6), gastrointestinal (I85.0, K25.0, K25.2, K25.4, K25.6, K26.0, K26.2, K26.4, K26.6, K27.0, K272, K27.4, K27.6, K28.0, K282, K28.4, K28.6, K29.0, K62.5, K92.0, K92.1, K92.2) and other major bleeding (D62, N02, R31, H11.3, H35.6, H43.1, H45.0, H92.2, J94.2, K66.1, M25.0, N92.0, N92.1, N92.4, N93.8, N93.9, N95.0, R04.0, R04.1, R04.2, R04.8, R04.9, R58, I312); and (3) death.

### 2.5. Baseline Characteristics of Individuals

We described demographic characteristics (age, sex) and specialty of the prescribers of oral ATs. Comorbidities were identified by hospital discharge/LTD diagnoses and specific procedures or drug reimbursements (see Appendix A). Because smoking status and alcohol consumption were not directly available from the databases, we used reimbursement of nicotine replacement therapy and hospital discharge diagnoses and related complications linked to tobacco use or alcohol heavy consumption (see Appendix A). Comedications were defined as drugs dispensed at least once within the 4 months before the index date. Polypharmacy was defined as 5 different drugs dispensed [18] at least once within the 4 months before the index date.

### 2.6. Statistical Analysis

The cumulative incidence curves of oral AT combinations were estimated with the Fine and Gray method [19], taking into account the competitive risk of death, first in the whole population and then in sub-groups defined a priori (<65, ≥65, ≥80 years old). We performed sensitivity analyses using different definitions of oral AT combinations (delivery of at least 2 different ATs for at least 30 or 45 successive days). We used the same approach to estimate the cumulative incidence of specific categories of oral AT combinations: dual antiplatelet therapies, dual therapies, triple therapies, appropriate and inappropriate oral AT combinations. The median maintenance of oral AT combinations was also estimated for all oral AT combinations and then for specific categories.

We estimated and compared the cumulative incidences of major bleeding according to the type of oral AT treatment initiated at study entry: oral AT monotherapy and oral AT combination, taking into account the competitive risk of death with the Fine and Gray model [19]. Follow-up was censored 7 days after the end of the covered period. Individuals were followed until the earliest of the following: death; major bleeding; diagnosis of auto-immune disease, hemophilia, HIV, or active cancer (see Appendix A); or end of the study period (31 December 2017).

Then, in patients initiating an oral AT combination (either at study entry or during follow-up), we compared the cumulative incidence of major bleeding since oral AT combination initiation according to the appropriateness of the oral AT combination.

We also estimated survival with the Kaplan-Meier method and compared survival rates according to the type of oral AT treatment initiated at study entry using Cox models. Models were adjusted on factors known to be associated with bleeding, death, or use of oral AT combinations in the literature, namely age, sex, renal and hepatic failure, vascular diseases (coronary heart disease, NV-AF, valvular heart disease, peripheral vascular disease, venous thromboembolism disease, stroke or arterial embolism, hypertension, diabetes) and history of major bleeding or anemia before the index date [16]. Results of the models are reported as hazard ratios (HRs) for Cox models and subdistribution HR (sHR) for Fine and Gray models, with 95% confidence intervals (CIs). All analyses involved using SAS Enterprise Guide v7.15. Two-tailed *p* < 0.05 was considered statistically significant.

## 3. Results

### 3.1. Baseline Characteristics of Individuals

During the study period, 22,220 individuals were included (Figure 1).

At study entry, 21,233 individuals initiated oral AT monotherapy and 987 an oral AT combination (576 an appropriate and 411 an inappropriate oral AT combination) (Figure 1). The mean (SD) age was 68 (12) years; 11,048 (50%) individuals were male. Baseline characteristics of individuals are described in Table 1. Individuals with an appropriate oral AT combination at study entry were more often younger and male and had coronary artery disease, whereas individuals with an inappropriate oral AT combination at study entry had more frequently indications for the use of oral anticoagulants (NV-AF and valvular heart disease) (Table 1). At study entry, 98% of appropriate oral AT combinations were dual antiplatelet therapy (*n* = 563) and 2% (*n* = 10) were dual therapy as compared with 83% (*n* = 340) and 16% (*n* = 66) of inappropriate oral AT combinations, respectively (see Appendix A).

### 3.2. AT Prescriptions

In all, 83% of all AT deliveries had been prescribed by general practitioners (GPs) (see Appendix A). GP prescriptions were mainly renewals. The most commonly delivered oral AT during the study period was aspirin, with 80% (*n* = 17,735) of individuals with at least one aspirin delivery, followed by clopidogrel (13%), rivaroxaban (10%), fluindione (7%) and apixaban (7%) (see Appendix A).

### 3.3. Cumulative Incidence of Oral AT Combination

The total number of oral AT combinations was 5945 during the study period. The median number of oral AT combinations per person, in the population with at least one prescription of oral AT combination, was 1 [25–75 interquartile range [IQR] 1–1] (range 1–10).

The cumulative incidence of oral AT combinations, considering the competitive risk of death, for the whole cohort over the 5-year study was 27.8% (95% CI 26.8 to 28.9) (Table 2, Figure 2). Sensitivity analyses are described in Appendix A: the longer the definition used to describe an oral AT combination, the more the cumulative incidence was reduced (cumulative incidence for oral AT combinations: 22.4%, 95% CI 21.6 to 23.3, and 18.6%, 95% CI 17.8 to 19.3, with 30 and 45 days, respectively, as cut-offs for definition). The prescriber of the first prescriptions of oral AT combinations was the hospital physician in 60% of cases, the GP in 17%, a cardiologist in 20% or other specialists in 3% (Appendix A). The most commonly prescribed oral AT combinations were dual antiplatelet therapies, with a cumulative incidence of 18.7% (95% CI 17.9 to 19.5) (especially before age 65) and dual therapies, with a cumulative incidence of 9.1% (8.3 to 9.9) (especially after age 65) (Table 2). Triple therapies were rare. The median maintenance of oral AT combinations was 115 days (25–75 IQR 30–360), longer for dual antiplatelet therapy (239 (53–412) days) than dual therapy (42 (57–135) days) or triple therapy (50 (25–107)days) (see Appendix A).

### 3.4. Appropriateness of Oral AT Combinations

When examining the appropriateness of all oral AT combinations (*n* = 5945), 370 (6%) were contraindicated and, among the other oral AT combinations, 3446 (58%) had no recommended indication for use: 51% for dual antiplatelet therapies, 92% for dual therapies and 69% for triple therapies (Figure 3). In total, 64% of oral AT combinations were inappropriate, in terms of indication, according to guidelines (Figure 3). Sensitivity analyses are presented in Appendix A: the longer the cut-off used to describe an oral AT combination, the lower the proportion of inappropriate oral AT combinations (proportion of oral AT combinations with non-recommended indication: 58% and 51% with 30 or 45 days used as cut-offs).

### 3.5. Risk of Major Bleeding and Death

The cumulative incidence of major bleeding, considering the competitive risk of death at 5 years, was 4.1% (95% CI 3.7 to 4.6). We compared the cumulative incidence of major bleeding according to the type of oral AT treatment initiated at study entry: oral AT monotherapy (*n* = 21,233) versus oral AT combination (*n* = 987), taking into account the competitive risk of death. Risk of major bleeding was higher for individuals with oral AT combination versus oral AT monotherapy (Table 3; sHR 2.16, 95% CI 1.01 to 4.63). In patients who started an oral AT combination, the cumulative incidence of major bleeding did not significantly differ according to the appropriateness of oral AT combinations (Appendix A).

The cumulative incidence of death over the entire study period was 10.8% (95% CI 10.1 to 11.6). The risk of death did not significantly differ according to the type of oral AT treatment initiated at study entry: oral AT monotherapy (*n* = 21,233) versus oral AT combination (*n* = 987), HR 1.42, 95% CI 0.65 to 3.14 (Table 4)

## 4. Discussion

In this large nationwide cohort study including 22,220 incident oral AT users ≥ 45 years old, oral AT combinations were frequent (cumulative incidence at 5 years: 27.8%, 95% CI 26.8 to 28.9) and often prescribed without indication complying with guidelines (64% of oral AT combinations). Risk of major bleeding increased for individuals initiating an oral AT combination versus an oral AT monotherapy (sHR 2.16, 95% CI 1.01 to 4.63).

### 4.1. Strengths and Weaknesses of the Study

To our knowledge, this is the first study to assess the incidence of oral AT combinations among incident oral AT users and to estimate the rate of inappropriate prescriptions in all common vascular diseases. Additionally, we used the EGB databases, covering a population representative of the French national population for age, sex, occupation, and medical expenses, which ensures the generalizability of the results at a national level. In addition, we could study this cohort of 22,220 individuals, followed for 5 years, since the first year of DOA deliveries in France. Finally, information on both oral AT deliveries and major bleedings or deaths were prospectively collected for an administrative purpose and thus independently of any pre-specified hypothesis.

Our study also has some limitations inherent to studies conducted with claims databases. First, exposure to treatment was based on claims for drug deliveries in pharmacies, which do not indicate how the individual actually takes the medications (most documented approach to reflect treatment) [3,4,11,12,20]. Otherwise, the cut-off for defining AT combinations, fixed at 15 days of combinations, may have led to overestimating the exposure to an AT combination (2 successive and close prescriptions with different ATs in monotherapy). However, sensitivity analyses, with longer thresholds at 30 and 45 days, gave concordant results. Second, to define the appropriateness of the oral AT combination prescription, we used only the indication for use, without taking into account the doses and durations, which probably underestimated the prevalence of inappropriate prescriptions. However, we may have overestimated this prevalence by missing some events leading to the delivery of oral AT combinations. Regardless, these events are easily identifiable in the EGB (hospitalization or medical procedures), and only recent events (<1 month) can lead to oral AT combination prescriptions. In addition, it may be appropriate to prescribe outside the guidelines for some individuals or contexts, but these situations are rare, specific [16,17], and most of them were excluded (requiring an expert opinion). Thirdly, we defined major bleeding as hospitalization with bleeding as a principal diagnosis or related diagnoses. This definition, usually used in studies using healthcare databases [3,21,22], is a pragmatic way to identify severe bleeding events from a patient perspective (bleeding requiring in-hospital care and/or with consequences leading to hospitalization). Because minor bleedings do not lead to hospitalization, they are not adequately identified in the EGB database and most of the time they do not require a change in antithrombotic treatment. For all these reasons, the bleeding rate may be underestimated.

### 4.2. Comparisons with Previous Studies

In accordance with the literature, GPs were the main prescribers of oral AT renewals [23] whereas hospital physicians and cardiologists were the main prescribers (75%) of oral AT combination initiations [9].

In all, 64% of all oral AT combinations used had no recommendations for use (6% of them being always contraindicated). However, this worrying result is lower than that previously assessed in a study involving clinical vignettes (76% of inappropriate prescriptions) [10], in which we also took into account the doses and durations of oral AT prescriptions to define appropriateness. The most frequent types of oral AT combinations out of recommendations we observed concerned dual and triple therapies, as was previously found in a European cohort of patients with atrial fibrillation [9]: 92% versus 95.3% for dual therapy with no indication and 69% versus 63.8% for triple therapy with no indication. Prolonging dual therapy, when oral anticoagulant monotherapy would be recommended, is particularly frequent and dangerous [9,16,17]. These inappropriate prescriptions, in connection with polypathology (especially the association of atrial fibrillation and coronary artery disease [16,17]), were most often found in individuals over age 80, those most at risk of severe bleeding.

As already known, we found that oral AT combinations increased the risk of major bleeding as compared with oral AT monotherapies [3,4]. However, we did not find a difference regarding the appropriateness of the prescription, which may be due to a lack of power. Indeed, inappropriate oral AT combinations comprised 16% of dual therapies (vs 2% for appropriate oral AT combinations), which are known to particularly increase the risk of bleeding [3,4]. The cumulative incidence of major bleeding was consistent with previous studies using the SNDS [21,22].

### 4.3. Implications for Clinical Practice and Public Health

Our results clearly reflect a gap between the production of recommendations and their implementation in daily clinical practice, with an impact on patient outcomes. We previously showed that international guidelines on oral AT combinations were numerous and frequently updated, and none encompassed all clinical situations [16]. Information dissemination could be an impediment for physicians to comply with guidelines. Health professionals and decision makers should increase educational efforts to achieve a better and more widespread implementation of current recommendations to improve the benefit/risk ratio of oral ATs. Several ways exist to help physicians prescribe according to the guidelines. Regular collaborations between specialists and GPs could be a first step for improvement. The lack of physicians, time, and resources is probably a barrier to this recommendation. We lack a digital prescription support tool focused on ATs, particularly oral AT combinations, adapted to clinical practice. Yet, computerized clinical decision support systems, providing assistance to clinicians in the process of decision making, have been shown to reduce medication errors and improve practitioner performance [24,25,26]. We are currently working on the development of such a digital prescription support tool whose real-life evaluation will begin in 2021.

## 5. Conclusions

In this large nationwide cohort study including 22,220 incident oral AT users aged ≥ 45 years old, 27.8% were exposed to an oral AT combination at least once within 5 years of oral AT initiation. Overall, 64% of oral AT combinations used had no recommended indication for use with an increased risk of major bleeding. People aged ≥ 80 years were most exposed to this inappropriate prescribing despite also being most at risk of bleeding.

## Figures and Tables

**Figure 1 jcm-10-02367-f001:**
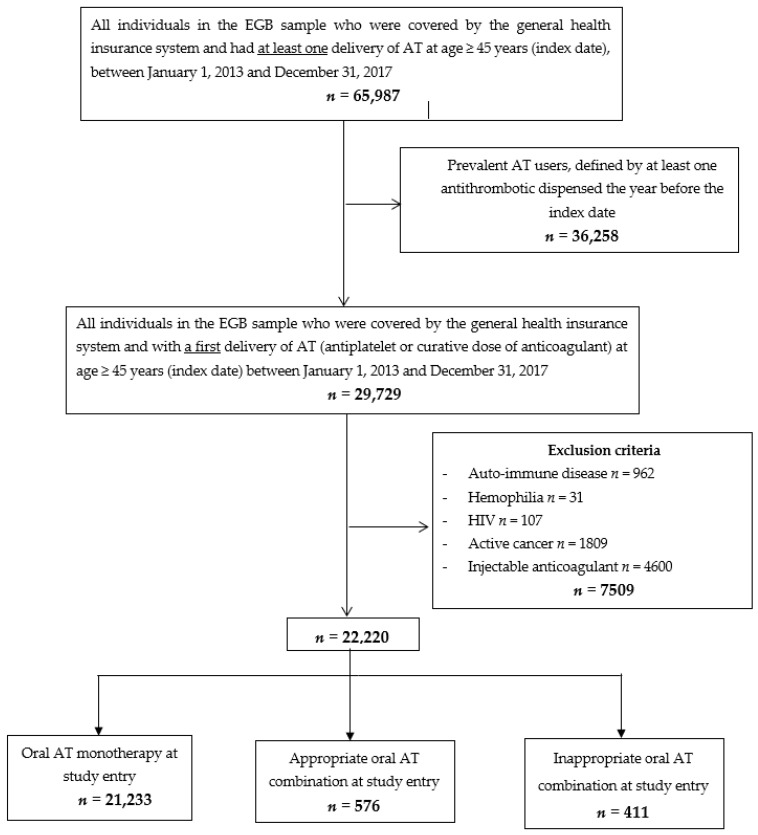
Flow chart. Abbreviations: EGB: échantillon généraliste des bénéficiaires (general sample of beneficiaries); AT: antithrombotics

**Figure 2 jcm-10-02367-f002:**
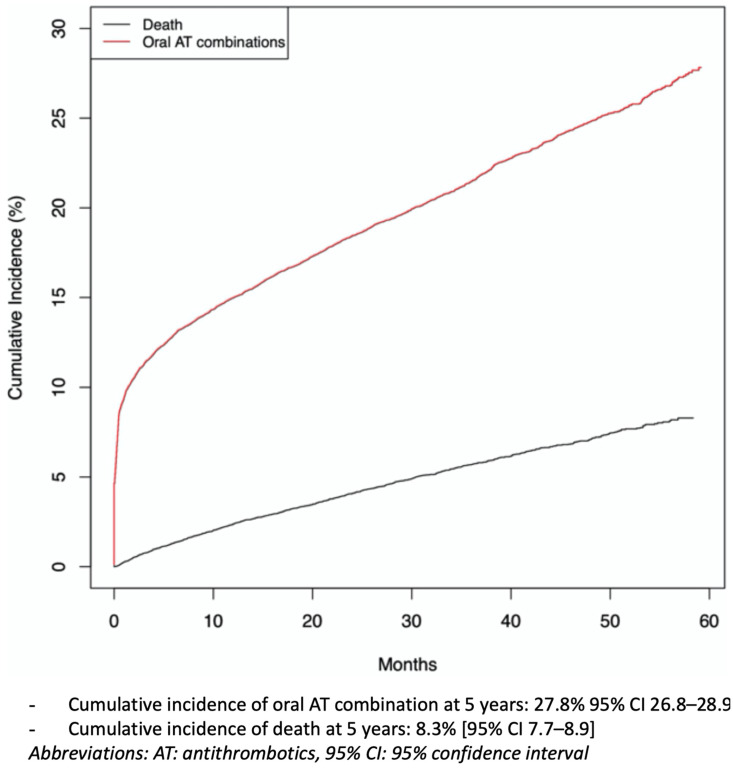
Cumulative incidence curves of oral antithrombotic (AT) combination and death for the whole cohort (*n* = 22,220), over the 5-year study.

**Figure 3 jcm-10-02367-f003:**
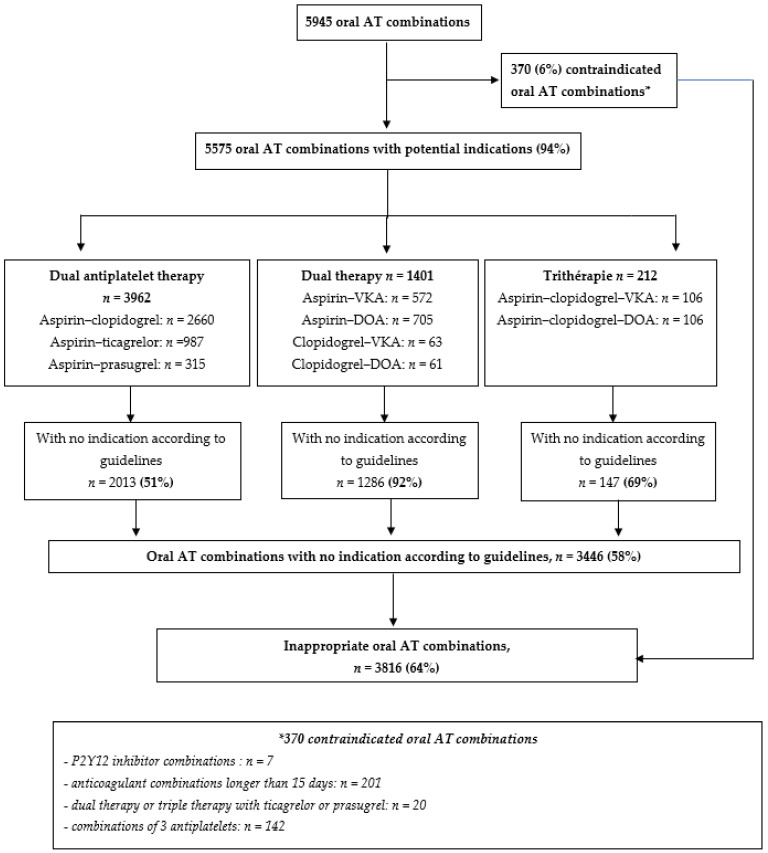
Proportion of inappropriate oral antithrombotic combination. Abbreviations: AT: antithrombotics; DOA: direct oral anticoagulant; VKA: vitamin K antagonist.

**Table 1 jcm-10-02367-t001:** Baseline characteristics of individuals: overall (*n* = 22,220), with oral antithrombotic (AT) monotherapy (*n* = 21,233), an appropriate oral AT combination (*n* = 576) or an inappropriate oral AT combination (*n* = 411) at study entry. Values are number (percentages) unless stated otherwise.

	Total *n* = 22,220	Oral AT Monotherapy *n* = 21,233	Appropriate Oral AT Combination *n* = 576	Inappropriate Oral AT Combination *n* = 411	*p*
Mean (SD) age, years	68 (12)	68 (12)	64 (12)	68 (12)	<0.0001
45–64	9310 (42)	8807 (42)	337 (58)	166 (40)	
65–79	8624 (39)	8295 (39)	166 (29)	163 (40)	
≥80	4286 (19)	4131 (19)	73 (13)	82 (20)	
Sex					<0.0001
Male	11,048 (50)	10,411 (49)	405 (70)	232 (56)	
Comorbidities					
Hypertension	12,862 (58)	12,311 (58)	300 (52)	251 (61)	0.008
Diabetes	5109 (23)	4942 (23)	78 (14)	89 (22)	<0.0001
Dyslipidemia	7243 (33)	6973 (33)	142 (25)	128 (31)	0.0002
Obesity	2206 (10)	2073 (10)	87 (15)	46 (11)	<0.0001
Coronary heart disease	3278 (15)	2578 (12)	568 (99)	132 (32)	<0.0001
Non-valvular atrial fibrillation	1935 (9)	1862 (9)	11 (2)	62 (15)	<0.0001
Valvular heart disease	487 (2)	402 (2)	7 (1)	78 (19)	<0.0001
Heart failure	766 (3)	721 (3)	16 (3)	29 (7)	0.0002
Peripheral vascular disease	1512 (7)	1424 (7)	42 (7)	46 (11)	0.001
VTE disease	759 (3)	739 (3)	10 (2)	10 (2)	0.04
Stroke or arterial embolism	1497 (7)	1456 (7)	6 (1)	35 (9)	<0.0001
Chronic kidney disease	818 (4)	773 (4)	19 (3)	26 (6)	0.01
Chronic hepatic disease	486 (2)	464 (2)	11 (2)	11 (3)	0.71
Anemia	1259 (6)	1203 (6)	24 (4)	32 (8)	0.05
History of bleeding	612 (3)	587 (3)	14 (2)	11 (3)	0.89
Dementia	781 (4)	767 (4)	6 (1)	8 (2)	0.0009
COPD	829 (4)	784 (4)	22 (4)	23 (6)	0.13
Smoking	952 (4)	906 (4)	23 (4)	23 (6)	0.39
Alcohol	946 (4)	902 (4)	30 (5)	14 (3)	0.37
Comedications					
Median (IQR) Number of drugs	9 (6–13)	9 (6–13)	10 (7–13)	10 (7–15)	<0.0001
Polypharmacy ^a^	18,865 (85)	17,932 (84)	557 (97)	376 (91)	<0.0001

Abbreviations: COPD: chronic obstructive pulmonary disease; IQR: 25–75 interquartile range; SD: standard deviation; VTE: venous thromboembolism ^a^ defined as 5 different drugs dispensed at least once within the 4 months before the index date.

**Table 2 jcm-10-02367-t002:** Cumulative incidence (%, 95% confidence interval) at 5 years of oral antithrombotic (AT) combinations considering the competitive risk of death (*n* = 1090), for the whole population (*n* = 22,220) and stratified by age.

	Total (*n* = 22,220)	<65 Years Old (*n* = 9310)	≥65 Years Old (*n* = 12,910)	≥80 Years Old (*n* = 4286)
Oral AT combinations (*n* = 4466)	27.8 (26.8–28.9)	28.1 (26.7–29.6)	27.8 (26.3–29.2)	24.8 (22.8–26.8)
Dual antiplatelet therapy (*n* = 3134)	18.7 (17.9–19.5)	22.3 (21.0–23.5)	16.2 (15.1–17.3)	13.5 (12.0–15.0)
Aspirin–clopidogrel (*n* = 2141)	13.7 (12.9–14.4)	14.2 (13.2–15.3)	13.4 (12.3–14.5)	11.9 (10.5–13.5)
Aspirin–ticagrelor (*n* = 754)	4.5 (4.1–4.9)	7.2 (6.3–8.1)	2.7 (2.3–3.0)	1.7 (1.3–2.3)
Aspirin–prasugrel (*n* = 239)	1.3 (0.01–1.5)	2.3 (1.9–2.7)	0.6 (0.5–0.8)	0.04 (0.001–0.2)
Dual therapy (*n* = 1075)	9.1 (8.3–9.9)	5.9 (5.0–7.0)	11.3 (10.1–12.6)	11.1 (9.3–12.9)
Aspirin–VKA (*n* = 409)	3.4 (2.9–4.1)	2.5 (1.8–3.3)	4.1 (3.2–5.1)	4.3 (3.6–5.2)
Aspirin–DOA (*n* = 587)	5.4 (4.8–5.9)	3.0 (2.4–3.7)	6.9 (6.0–7.9)	6.5 (5.0–8.4)
Clopidogrel–VKA (*n* = 32)	0.2 (0.1–0.3)	0.2 (0.08–0.3)	0.2 (0.1–0.4)	0.2 (0.08–0.5)
Clopidogrel–DOA (*n* = 47)	0.4 (0.3–0.6)	0.3 (0.2–0.6)	0.5 (0.3–0.7)	0.4 (0.2–0.7)
Triple therapy (*n* = 67)	0.8 (0.5–1.3)	0.5 (0.2–1.1)	1.0 (0.6–1.8)	0.7 (0.4–1.3)
Aspirin–clopidogrel–VKA (*n* = 23)	0.3 (0.1–0.5)	0.3 (0.06–1.0)	0.2 (0.1–0.4)	0.4 (0.1–0.9)
Aspirin–clopidogrel–DOA (*n* = 44)	0.6 (0.3–1.0)	0.2 (0.1–0.4)	0.8 (0.4–1.7)	0.4 (0.2–0.7)
Appropriate ^a^ oral AT combinations (*n* = 1879)	11.1 (10.4–11.8)	14.1 (13.0–15.1)	9.1 (8.2–10.1)	7.1 (6.0–8.3)
Inappropriate ^a^ oral AT combinations (*n* = 2587)	18.9 (17.9–19.8)	16.3 (15.0–17.6)	20.6 (19.2–22.0)	19.1 (17.2–21.1)

For cumulative incidence, only the first oral AT combination of interest per person is used. Abbreviations: DOA: direct oral anticoagulant; VKA: vitamin K antagonist. ^a^ Appropriate or inappropriate oral AT combinations: according to guidelines.

**Table 3 jcm-10-02367-t003:** Risk of major bleeding in individuals starting the study with oral antithrombotic (AT) combinations (*n* = 987) versus individuals starting the study with oral AT monotherapy (*n* = 21,233) as a reference (estimated by the fitted Fine and Gray model with death as competitive event).

Risk of Major Bleeding ^$^ Variables (Number in the Class)	sHR (95% CI)	*p* Value
Oral AT combinations at study entry ^a^ (*n* = 987)	2.16 (1.01–4.63)	0.048
Male (*n* = 11,048)	1.20 (0.77–1.86)	0.42
Age at study entry, years		0.007
Age ^b^ 65–79 (*n* = 8624)	1.55 (0.90–2.68)	
Age ^b^ ≥ 80 (*n* = 4286)	2.53 (1.41–4.53)	
Chronic kidney disease (*n* = 818)	2.83 (1.41–5.70)	0.003
Chronic hepatic disease (*n* = 486)	2.37 (0.95–5.93)	0.06
Coronary heart disease (*n* = 3278)	1.12 (0.65–2.14)	0.59
Peripheral vascular disease (*n* = 1512)	0.87 (0.41–1.85)	0.72
Non-valvular atrial fibrillation (*n* = 1935)	1.45 (0.81–2.59)	0.21
Valvular heart disease (*n* = 487)	1.10 (0.40–3.03)	0.85
Stroke or arterial embolism (*n* = 1497)	1.72 (0.92–3.22)	0.09
Venous thromboembolism disease (*n* = 759)	1.53 (0.66–3.59)	0.32
Hypertension (*n* = 12,862)	0.81 (0.51–1.29)	0.39
Diabetes (*n* = 5109)	0.62 (0.34–1.11)	0.11
Anemia (*n* = 1259)	1.48 (0.70–3.10)	0.30
History of major bleeding (*n* = 612)	1.98 (0.81–4.85)	0.14

^$^ Event of interest: hospitalization for major bleeding, *n* = 86; Event in competition: death = *n* = 167; ^a^ Reference group is oral AT monotherapy; ^b^ Reference: 45–64 years old; Abbreviations: sHR: subdistribution hazard ratio; 95% CI: 95% confidence interval.

**Table 4 jcm-10-02367-t004:** Risk of death in individuals starting the study with oral antithrombotic (AT) combinations (*n* = 987) versus individuals starting the study with oral AT monotherapy (*n* = 21,233) as a reference (estimated by the Fitted Cox model).

Risk of Death ^$^ Variables (Number in the Class)	HR (95% CI)	*p* Value
Oral AT combinations at study entry ^a^ (*n* = 987)	1.42 (0.65–3.14)	0.38
Male (*n* = 11,048)	1.33 (0.97–1.83)	0.08
Age at study entry, years		<0.0001
Age ^b^ 65–79 (*n* = 8624)	1.64 (0.97–2.77)	
Age ^b^ ≥ 80 (*n* = 4286)	9.25 (5.75–14.89)	
Chronic kidney disease (*n* = 818)	1.27 (0.72–2.22)	0.41
Chronic hepatic disease (*n* = 486)	2.57 (1.24–5.30)	0.01
Coronary heart disease (*n* = 3278)	0.80 (0.50–1.31)	0.38
Peripheral vascular disease (*n* = 1512)	1.41 (0.88–2.62)	0.14
Non-valvular atrial fibrillation (*n* = 1935)	1.37 (0.92–2.05)	0.12
Valvular heart disease (*n* = 487)	0.80 (0.32–1.98)	0.62
Stroke or arterial embolism (*n* = 1497)	1.27 (0.80–2.02)	0.31
Venous thromboembolism disease (*n* = 759)	2.37 (1.43–3.98)	0.0009
Hypertension (*n* = 12,862)	0.95 (0.68–1.33)	0.77
Diabetes (*n* = 5109)	1.04 (0.72–1.52)	0.83
Anemia (*n* = 1259)	1.58 (0.99–2.53)	0.05
History of major bleeding (*n* = 612)	1.46 (0.73–2.89)	0.28

**^$^** Event of interest: death, *n* = 167; ^a^ Reference group is oral AT monotherapy; ^b^ Reference group: 45–64 years old; Abbreviations: AT: antithrombotics; HR: hazard ratio; 95% CI: 95% confidence interval.

## Data Availability

Permanent access to the French healthcare databases is automatically granted to certain government agencies, public institutions, and public service authorities. Temporary access for studies and research is possible upon request from the National Health Data Institute (INDS). All databases used in this study contained only anonymous patient records. Publicly sharing EGB data is forbidden by law according to The French national data protection agency (Commission Nationale de l’Informatique et des Libertés, CNIL). To request data access, please contact The National Institute for Health Data (Institut National des Données de Santé, INDS; website: https://documentation-snds.health-data-hub.fr/ (accessed date 25 May 2021).

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
