# Peer review of "Inappropriate Use of Oral Antithrombotic Combinations in an Outpatient Setting and Associated Risks: A French Nationwide Cohort Study"

_jcm, 2021, doi:10.3390/jcm10112367_

Round 1

Reviewer 1 Report

The authors described nationwide 22220 incident oral AT users with 27.8% cumulative AT initiation within 5 years. Furthermore, there are inappropriate indication of AT combination in majority of those cases. It is an interesting topic which is highly relevant in our clinical practice. However, it is a national wide survey with its limitations: 

First, major bleeding is only defined by ICD Code I 61.- I think it is hard to define major bleeding with those ICD Codes since in my point of view, major bleeding has the definition of either indication of surgical treatment or clinical deterioration due to the bleeding. Solely with ICD Code, major and minor bleeding are not distinguishable. Therefore, 4% of major bleeding rate might be estimated false positively high. Maybe the authors could add OPS analysis to evaluate patients who needed surgical treatment in case of major bleeding. 

Second, the mortality rate was not different between combined AT vs mono-AT, which smoothen the problem of inappropriate indication of combined AT-treatment. It is hard to define the inappropriate indication, because there are lot of off-label use of AT for example NOACs in case of sinus venous thrombosis, low heparin treatment as a substitute of antithrombotic medication etc.  Therefore I personally don't think that 64% with no recommended use is a problem. The most important predictor for death, bleeding was the age and other comorbidities. 

Overall, it is a well written paper which had interesting aspect in use of AT. The lack of the paper might be the novelty of the main conclusion since age general is a risk for intracranial bleeding and usually the over treatment through an overdosage is mainly the problem in the clinical practice.

Reviewer 2 Report

This in an interesting paper regarding large nationwide cohort study including 22220 incident oral AT users showing as oral AT combinations were frequent (incidence at 5 years 27.8%) and often prescribed without indication complying with guidelines (64%). The risk of hospitalization for major bleeding was increased for individuals initiating an oral AT combination versus an oral AT monotherapy (sHR 2.16) Despite the numerous limitations already correctly stated by the authors in their discussion, I believe that the work cannot be improved further. However, the final information remains very clear and useful for changing our clinical practice. Furthermore, the large amount of data reported in the supplementary tables is very appreciated, which significantly enrich the whole manuscript.

Round 2

Reviewer 1 Report

The authors answered all comments appropriately.